# Combined Inhibition of Indolamine-2,3-Dioxygenase 1 and C-X-C Chemokine Receptor Type 2 Exerts Antitumor Effects in a Preclinical Model of Cervical Cancer

**DOI:** 10.3390/biomedicines11082280

**Published:** 2023-08-16

**Authors:** Solangy Lizcano-Meneses, Rogelio Hernández-Pando, Ian García-Aguirre, José Bonilla-Delgado, Víctor Manuel Alvarado-Castro, Bulmaro Cisneros, Patricio Gariglio, Enoc Mariano Cortés-Malagón

**Affiliations:** 1Department of Genetics and Molecular Biology, Centro de Investigación y de Estudios Avanzados del Instituto Politécnico Nacional (CINVESTAV-IPN), Mexico City 07360, Mexico; solangy.lizcano@cinvestav.mx (S.L.-M.);; 2Department of Pathology, Instituto Nacional de Ciencias Médicas y Nutrición “Salvador Zubirán”, Mexico City 14080, Mexico; 3Departamento de Bioingeniería, Escuela de Ingeniería y Ciencias, Instituto Tecnologico y de Estudios Superiores de Monterrey, Ciudad de México, Mexico City 14380, Mexico; 4Departamento de Bioingeniería, Escuela de Ingeniería y Ciencias, Instituto Tecnologico y de Estudios Superiores de Monterrey, Toluca 50110, Mexico; 5Research Unit, Hospital Regional de Alta Especialidad de Ixtapaluca, Ixtapaluca 56530, Mexico; 6Centro de Investigación de Enfermedades Tropicales, Universidad Autónoma de Guerrero, Acapulco 39640, Mexico; 7Research Division, Hospital Juárez de México, Mexico City 07760, Mexico; 8Genetics Laboratory, Hospital Nacional Homeopático, Mexico City 06800, Mexico

**Keywords:** immunotherapy, IDO-1, CXCR-2, K14E7, cervical cancer

## Abstract

Cervical cancer is a public health problem diagnosed in advanced stages, and its main risk factor is persistent high-risk human papillomavirus infection. Today, it is necessary to study new treatment strategies, such as immunotherapy, that use different targets of the tumor microenvironment. In this study, the K14E7E2 mouse was used as a cervical cancer model to evaluate the inhibition of indolamine-2,3-dioxygenase 1 (IDO-1) and C-X-C chemokine receptor type 2 (CXCR-2) as potential anti-tumor targets. DL-1MT and SB225002 were administered for 30 days in two regimens (R1 and R2) based on combination and single therapy approaches to inhibit IDO-1 and CXCR-2, respectively. Subsequently, the reproductive tracts were resected and analyzed to determine the tumor areas, and IHCs were performed to assess proliferation, apoptosis, and CD8 cellular infiltration. Our results revealed that combined inhibition of IDO-1 and CXCR-2 significantly reduces the areas of cervical tumors (from 196.0 mm^2^ to 58.24 mm^2^ in R1 and 149.6 mm^2^ to 52.65 mm^2^ in R2), accompanied by regions of moderate dysplasia, decreased papillae, and reduced inflammation. Furthermore, the proliferation diminished, and apoptosis and intra-tumoral CD8 T cells increased. In conclusion, the combined inhibition of IDO-1 and CXCR-2 is helpful in the antitumor response against preclinical cervical cancer.

## 1. Introduction

Cervical cancer (CC) is a public health problem in low- and middle-income countries [1]. This disease is strongly associated with persistent high-risk human papillomavirus (HR-HPV) infection. Although the prevalence of genital HR-HPV infections is relatively high in all populations, cervical cancer is less frequent than infection rates, suggesting that other factors that add to infection are necessary for malignant transformation [2]. Numerous studies have indicated that the tumor immune microenvironment (TIME) is an essential factor for the development, persistence, and even radio and chemical resistance of CC [3,4,5]. Immunosuppression is a crucial characteristic in TIME and is characterized by the presence of regulatory T cells (Tregs), myeloid-derived suppressor cells (MDSCs), tumor-associated neutrophils (TAN), tumor-associated macrophages (TAMs), and amino acid-derived metabolites [6,7].

Indolamine-2,3-dioxygenase 1 (IDO-1) is the main enzyme responsible for the catabolism of tryptophan (Trp) in the kynurenine pathway (Kyn), leading to a local decrease in Trp and the production of immunosuppressive metabolites [8]. It is expressed in immature dendritic cells, secondary lymphoid organs and epithelial cells of the female genital tract, placenta endothelial cells, the lung, interstitial cells of the intestinal tract, and β-pancreatic cells [9]. Moreover, constitutive expression of this enzyme has also been reported in approximately 60% of human tumors [10]. Concerning CC, an increase in IDO-1 positive squamous cells and a higher Kyn/Trp ratio have been associated with immune suppression, clinicopathological parameters, and poor survival [11,12,13,14]. Furthermore, IDO-1 has been reported to be involved in the self-renewal and expression of *OCT4* and *SOX2* in cervical cancer stem cells [15].

Another protumor factor for TIME is the deregulation of the CXCR-2/CXCLs axis. CXCR-2 is a receptor for CXCL1–3 and CXCL5–8 chemokines, which are known for their ability to recruit immune cells. Upon binding to its ligands, CXCR-2 triggers robust neutrophil, macrophage, and MDSC chemotaxis [16]. Additionally, the CXCR-2/CXCL axis plays a role in tumorigenesis, angiogenesis, and metastasis [17,18,19,20,21,22,23]. In the case of CC, overexpression of CXCL1 and CXCR2 has been reported [24]. Furthermore, the administration of exogenous CXCL3 or CXCL5 on cervical cancer cell lines contributes to proliferation and migration [25,26,27]. Moreover, using an in vivo model of cervical cancer, our research group and others have demonstrated the expression of a set of chemokine genes (*Cxcl1*, *Cxcl2*, *Cxcl3*, and *Cxcl5*) in stromal and epithelial cancer cells, revealing biological crosstalk in cervical carcinogenesis [3,28].

Immunotherapy targeting TIME has emerged as a promising approach. In the context of CC, the FDA has approved using pembrolizumab in patients who test positive for programmed cell death ligand 1 (PD-L1). However, the response rate to this treatment has been relatively low, ranging from 10% to 17% [29,30]. Single-agent immunotherapies, exemplified by agents like pembrolizumab in the context of CC, may encounter limitations due to tumor heterogeneity, immune escape mechanisms, or microenvironmental factors [31]. Combining different immunotherapeutic agents, such as immune checkpoint inhibitors, adoptive T cell therapies, cytokines, and vaccines, can create a more comprehensive and dynamic immune response against the disease. For example, studies indicated that when used as a single agent, IDO-1 or CXCR-2 inhibitors exerted little antitumor efficacy, while combination with other therapies showed markedly enhanced antitumor effectiveness [32,33,34,35]. On the contrary, a phase 1 clinical study yielded inconclusive evidence concerning the potential benefits of combined therapy incorporating the IDO-1 inhibitor (navoximod) and the PD-L1 inhibitor (atezolizumab) in solid tumors, including CC [36]. Therefore, exploring the effects of the IDO-1 inhibitor (1-DL-MT) and the CXCR-2 inhibitor (SB225002) can be useful based on their distinct mechanisms of action, which hold the potential for a complementary impact on specific pathways crucial to the progression of CC. Our study aimed to investigate the influence of inhibiting IDO-1 and CXCR-2 on cervical tumor growth. To achieve this, we utilized pharmacological inhibitors in single-drug or combination approaches using a preclinical model of cervical cancer (K14E7E2).

## 2. Materials and Methods

### 2.1. Cervical Cancer Mouse Model (K14E7E2)

In this study, we utilized the K14E7 mouse model. The K14E7 transgenic mouse has been demonstrated to develop lesions resembling cervical intraepithelial neoplasia I (CIN-I) after 1 month of estradiol treatment, CIN-II after 3 months, and high-grade dysplastic lesions and invasive cancer (CIN-III and CIS) when subjected to continuous estrogen treatment for 6 months [28,37]. A 6-month treatment with 17-beta estradiol (E2) (Cat# SE-121; Innovative Research of America; Sarasota, FL, USA) was administered to induce cervical cancer in the mice for our study. Briefly, 1-month-old K14E7 transgenic mice were anesthetized with isoflurane using the IMPAC^6^ (VetEquip Inhalation Anesthesia Systems; Livermore, CA, USA) and a sustained release pellet that delivered 0.05 mg/60 days of E2 was implanted on the dorsal skin. The pellets were inserted every 2 months until mice reached 7 months of age. Finally, for this study, this model was designated as K14E7E2.

The study was conducted under the guidelines established by the Laboratory Animal Production and Experimentation Unit (UPEAL-CINVESTAV-IPN, Mexico; NOM-451 062-ZOO-1999) and received approval on 8 October 2019, from the internal committee responsible for care and use of the laboratory animals at the Centro de Investigación y de Estudios Avanzados del Instituto Politécnico Nacional (protocol code: 0295-19).

### 2.2. Inhibition Therapy

For inhibition therapy, the 1-methyl-DL-tryptophan (1-DL-MT, IDO-1 inhibitor) (Cat# 860646; Sigma-Aldrich; Darmstadt, Hesse, Germany) was prepared as 20 mM stock in 0.1 N NaOH (pH 7.4) (Cat# S5881; Sigma-Aldrich; Darmstadt, Hesse, Germany). The stock solution was then diluted in drinking water at a 2 mg/mL concentration and supplemented with 2 g/L of aspartame (Equal; New Providence, NJ, USA). To ascertain the average daily water consumption of the mice, we calculated the difference in solution volume and subsequently divided it by the number of days and the number of mice in each cage. The mice consumed an average of 5 mL/day, and the water was replaced every 5 days [38]. SB225002 (CXCR-2 inhibitor) (Cat# SML071; Sigma-Aldrich; Darmstadt, Hesse, Germany) was prepared as 50 mM stock in DMSO (Cat# D8418; Sigma-Aldrich; Darmstadt, Hesse, Germany) and diluted in 10% DMSO with 1× PBS, and mice were administered intraperitoneally with 2.5 mg/kg once/2 days of SB225002 [39].

Due to exogenous estradiol dependency for the continued growth of cervical tumors [40], the K14E7E2 mice were divided into two treatment regimens: In the first regimen (R1), 6-month-old mice were administered a combination of 1-DL-MT and SB225002 for 30 days before completing the treatment with E2. These mice were then euthanized at 7 months old. In the second regimen (R2), 7-month-old mice, after completing the treatment of E2, were treated with a combination of 1-DL-MT and SB225002 for 30 days and euthanized at 8 months old. In both regimens, the efficacy of monotherapy with 1-DL-MT or SB225002 was also evaluated. The K14E7E2 mice treated with vehicle solutions [0.1 N NaOH diluted in drinking water (pH 7.4) supplemented with 2 g/L of aspartame and 10% DMSO diluted in 1× PBS] served as the group that developed CIN-III/CC.

All mice were anesthetized using isoflurane with the IMPAC^6^ (VetEquip Inhalation Anesthesia Systems; Livermore, CA, USA) and then sacrificed through cervical dislocation. The female reproductive tracts were removed, and the endocervix and ectocervix zone was dissected. The dissected tissue was fixed in 4% paraformaldehyde (Cat# 158127 Sigma-Aldrich; Darmstadt, Hesse, Germany) overnight at 4 °C, embedded in paraffin (Cat# 76242; Sigma-Aldrich; Darmstadt, Hesse, Germany), or stored at −70 °C for future experiments. Four to six mice were used from each experimental procedure from each group.

### 2.3. Western Blot

The entire reproductive tracts of three K14E7 and three K14E7E2 mice were harvested. The zone containing the endocervix and the ectocervix was dissected and macerated using liquid nitrogen. The obtained samples were added to tubes containing 1 mL of T-PER buffer lysis (Cat#. 78510; Thermo Fisher Scientific, Waltham, MA USA) and halt protease inhibitors cocktail (Cat# 78430; Thermo Fisher Scientific, Waltham, MA USA), then centrifuged at 10,000× *g* for 15 min at 4 °C. The supernatants were collected, and the protein concentration was determined using the Bradford assay (BioRad, Hercules, CA, USA). Next, protein samples were prepared using 2× Laemmli Sample Buffer (Cat# 1610737; BioRad, Hercules, CA, USA) and heated at 95 °C for 5 min. Next, 40 μg of protein sample mix was loaded onto a 10% polyacrylamide gel, and electrophoresis was performed at 100 V for 1 h using Mini-PROTEAN Tetra Vertical Electrophoresis Cell (BioRad, Hercules, CA, USA). Subsequently, electrophoretic protein transfer to a nitrocellulose membrane (0.45 µm pore-size) was achieved at 20 V for 40 min using the Trans-Blot semi-dry transfer cell (BioRad, Hercules, CA, USA). The membranes were incubated with a blocking buffer (1× TBST with 3% *w*/*v* of nonfat dry milk) for 1 h at RT. IDO-1 rabbit mAb (3:1000) (Cat# 51851S; Cell Signaling Technology, Danvers, MA, USA) was added to the membrane and incubated overnight at 4 °C, followed by washing the membranes three times for 5 min in 1× TTBS. Afterward, the membranes were incubated with an anti-rabbit IgG antibody linked to HRP (1:3000) (Cat# 7074S; Cell Signaling Technology, Danvers, MA, USA) for 1 h at RT. The membranes were washed three times for 5 min in 1× TTBS, and the chemiluminescence reaction was performed using the Clarity Western ECL Substrate (BioRad, Hercules, CA, USA). The images were captured using the C-DIGIT blot scanner (LI-COR Biosciences, Lincoln, NE, USA). The amount of protein was determined with normalization using the fixed-point method, utilizing the Image Studio software (LI-COR Biosciences, Lincoln, NE, USA). The Anti-GAPDH antibody (1:1000) (Cat# 32233; Santa Cruz Biotechnology, Dallas, TX, USA) was used to detect the internal reference protein. Total placenta extract was used as the positive control for IDO-1.

### 2.4. Histopathology and Tumor Area

The paraffin-embedded tissues previously described were serially sectioned into 5 μm thick slices using the HistoCore BioCut microtome (Leica Biosystems; Deer Park, IL, USA). A total of 10 sections were collected at 75 μm intervals for hematoxylin and eosin staining and immunohistochemistry. Briefly, the tissue paraffin sections were incubated at 56 °C for 1 h and then immersed in xylene twice for 5 min each. Subsequently, the tissue sections were washed in a decreasing alcohol gradient (100%, 90%, and 70%) and rinsed with 1× PBS. The tissue sections were stained with hematoxylin and eosin and histopathologically classified to determine the grade of the cervical lesion present and the size of the cancers in each animal, as previously described [37]. Cervical intraepithelial neoplasia III was classified as tissues characterized by cells with increased nucleus size, a high degree of anaplasia, an increased frequency, and distribution of dysplastic cells in the suprabasal layers of the squamous epithelium, with projection into the cervical stroma. Cancer in situ (CIS) was classified as tissue containing abundant anaplastic cells with significantly increased nuclear size and a pronounced degree of remodeling and undulation of the epithelial-stromal border while retaining an intact basement membrane without evidence of microinvasion. Invasive, well-differentiated squamous cancers comprised dysplastic cell invasion through the basement membrane. An expert pathologist performed the histopathology.

The hematoxylin- and eosin-stained images were captured using the Zeiss Axio Imager A2 microscope (Carl Zeiss Microscopy, White Plains, NY, USA). Tumor area analysis was conducted by calibrating the software through “the analyze and set scale” option at a known distance of 100 µm. The tumor area was delimited using the polygon selection tool and analyzed using the ROI manager tool. All analyses were performed using ImageJ 1.46 J software (https://imagej.nih.gov/ij/download.html, accessed on 10 October 2022). Three tissue sections from each of the six mice in each study group were selected, and three to five fields from each tissue section were analyzed.

### 2.5. Immunochemistry

For immunohistochemical staining, the tissue sections on the slide were initially deparaffinized and hydrated, as mentioned earlier. Antigen retrieval was then performed using 1× immunoDNA retriever citrate or 1× immunoDNA retriever with EDTA (Cat# BSB 0022 and BSB 0032; BioSB system; Santa Barbara, CA, USA), depending on antibody specifications. The retriever solution was heated to 110 °C for 10 min in a domestic pressure cooker (Cuisinart; Stamford, CT, USA). Subsequently, the tissue sections were tempered in 1× PBS at RT and incubated with a polydetector peroxidase blocker (Cat# BSB 0050; BioSB system; Santa Barbara, CA, USA) for 5 min. In the background blocking step, an ImmunoDNA background blocker (Cat# BSB 0107; BioSB system; Santa Barbara, CA, USA) was used for 10 min. The tissue sections were incubated overnight at 4 °C with primary antibodies against MCM2 (1:800), IDO-1 (1:800) (Cat# 3619S and 51851S, respectively; Cell Signaling Technology, Danvers, MA, USA); CXCL5 (1:100) (Cat# BS-2549R; Bioss Antibodies, Woburn, MA, USA); CD8 (1:100), PCNA (1:100) (Cat# SC-25280 and SC-1177; respectively; Santa Cruz Biotechnology, Dallas, TX, USA); MPO (1:8000) and Granzyme B (1:100) (Cat# ab188211 and ab4059; respectively; Abcam, Waltham, MA, USA). Afterward, the tissue sections were washed with 1× PBS and incubated with an HRP-linked secondary antibody (1:1000) (Cat# 7074S; Cell Signaling Technology, Danvers, MA, USA) for 1 h at RT. Protein detection was performed using the DAB chromogen. Finally, the tissue sections were stained with hematoxylin for 1 min and covered with Entellan (Cat# 1079600500; Sigma-Aldrich, Darmstadt, Hesse, Germany) mounting medium for microscopy. All experiments were performed with four mice from each group.

Immunohistochemistry-based quantifications were performed by capturing five fields within the cervical cancer tissue of each group. Brown pixels within the threshold were selected and quantified as positive cells for each field using Image Pro Plus 4.5.0.19. Software (Media Cybernetics, Rockville, MD, USA). Tissue sections without primary antibodies were included as a negative control.

### 2.6. Indoleamine 2,3-Dioxygenase 1 (IDO-1) Activity

To measure IDO-1 activity in mammalian tissues, we used the IDO1 Activity Assay Kit (Cat# Ab235936; Abcam; Waltham, MA, USA) according to the manufacturer’s instructions. Briefly, we harvested complete female reproductive tracts, and the entire zone containing the endocervix and ectocervix was isolated, macerated with liquid nitrogen, and homogenized in 500 μL IDO-1 ice-cold assay buffer, followed by centrifugation (10,000× *g*, 15 min, 4 °C), and then the supernatant was collected. After that, 50 μL reaction premix (2×) was prepared, mixed with 15 μL of the test sample, and made up to 90 μL with IDO1 assay buffer. Next, 10 μL of the 1 mM IDO-1 substrate solution was added to each assay well, and the plate was incubated at 37 °C in a dark environment for 45 min. Then 50 μL of the fluorogenic developer solution was added, the plate was incubated at 45 °C for 3 h, and the fluorescence (Ex/Em = 402/488 nm) was measured.

### 2.7. TUNEL Assay

According to the manufacturer’s instructions, the procedure was performed using the HRP-DAB tunel assay kit (Cat# Ab206386; Abcam, Waltham, MA, USA). Briefly, the tissue sections were deparaffinized for 1 h at 56 °C and washed twice in xylene for 5 min. Then, they were hydrated in a decreasing alcohol gradient (100%, 90%, and 70%, each for 5 min) and rinsed with 1× PBS. Next, 100 µL of peroxidase K solution was added, incubated for 20 min, and washed with 1× TBS; then, 100 µL of peroxidase block was added for 10 min and washed with 1× TBS. The slides were covered with 100 µL of TdT equilibration buffer and incubated for 30 min at room temperature. Following this, 40 µL of the TdT labeling reaction mix was added. Coverslips were added to the slides and then incubated for 90 min at 37 °C in a humid chamber. Next, the slides were washed with 1× TBS, and 100 µL of stop buffer was added and incubated for 5 min. Then 100 µL of blocking buffer was added and incubated for 10 min; the excess was removed, and 100 µL of the conjugate was added and incubated for 30 min in a humid chamber. It was washed with 1× PBS, 100 µL of DAB solution was added, and stained with 100 µL of methyl green for 3 min. Finally, the slides were covered with Entellan (Cat# 1079600500; Sigma-Aldrich, Darmstadt, Hesse, Germany) mounting medium for microscopy.

### 2.8. Analysis of the IDO1 and CXCLs Genes Using the GEPIA2 Database

To investigate the mRNA expression of *IDO1* and *CXCLs* genes in cervical cancer, we used the GEPIA2 platform (http://gepia2.cancer-pku.cn/#index, accessed on 8 December 2022). The mRNA levels of cervical squamous cell carcinoma/endocervical adenocarcinoma and normal cervical/endocervical tissue were compared. The statistical analysis was performed using Student’s *t*-tests, and Log2FC Cutoff > 1 and *p* < 0.01 were assumed significant. In addition, a Kaplan–Meier curve was used to analyze overall survival.

### 2.9. Statistical Analysis

The Shapiro–Wilk and Levene’s tests were conducted to assess the normal distribution and homogeneity of the variance for all the groups under study. In the case of skewed distribution, non-parametric tests were employed. We utilized the Kruskal–Wallis test, followed by Dunn, as a post hoc analysis for multiple pairwise comparisons. Parametric tests were used for the normally distributed data, including a one-way ANOVA with the Tukey test for multiple pairwise comparisons. The statistical analyses were performed using GraphPad Prism 9.5.1 software (GraphPad; San Diego, CA, USA).

## 3. Results

### 3.1. Targeting IDO-1 and CXCR-2 Inhibits Cervical Cancer

IDO-1 and the CXCR-2/CXCL axis are critical factors in the development of many solid tumors. Our research group previously reported the overexpression of *Ido1*, *Cxcl1*, and *Cxcl5* mRNAs in cervical malignant lesions in the K14E7E2 mouse model [28]. Consequently, the present study verified the expression of IDO-1 and CXCL5 at the protein level (Appendix A). Therefore, IDO-1 and CXCR-2 were proposed as therapeutic targets in CC using the mouse model K14E7E2. We established two treatment regimens, R1 and R2 (Figure 1A), as described in the Materials and Methods Section. Our study observed a notable and statistically significant decrease in tumor areas following combined therapy (CT) administration. In the R1 group, the tumor area decreased from 196.0 mm^2^ ± 26.76 mm^2^ SD to 58.24 mm^2^ ± 3.11 mm^2^ SD. Similarly, in the R2 group, the tumor area decreased from 149.6 mm^2^ ± 53.98 mm^2^ SD to 52.65 mm^2^ ± 10.12 mm^2^ SD. Notably, this decrease in tumor area remained consistent and significant regardless of the dependency on exogenous estradiol.

Moreover, we observed accompanying changes in the affected regions, including moderate dysplasia, papillae reduction, and inflammation, compared to the group that did not receive combined therapy (Figure 1B, panel K14E7E2). To further evaluate the effectiveness of the treatment, we also investigated the use of similar regimens as monotherapy. Specifically, we examined the effects of the monotherapies shown in Figure 1B, panels K14E7E2+IDO-1i and K14E7E2+CXCR-2i. While we observed a decrease in tumor areas with these monotherapy approaches, especially in the K14E7E2+IDO-1i group, no statistically significant effects were found compared to the K14E7E2 group, as demonstrated in Figure 1C.

Considering that the combined inhibition of IDO-1 and CXCR-2 (K14E7E2 + CT) exhibited a more significant effect in reducing cervical tumor areas compared to single-drug treatment in mice that developed CIN-III/CC (K14E7E2), we decided to utilize this regimen for our subsequent experiments. We investigated the influence of CT on IDO-1 expression and activity, as well as neutrophil recruitment using myeloperoxidase (MPO) staining. As shown in Figure 2A, IDO-1 expression was primarily observed in cellular infiltration within cervical cancer samples. However, the immunohistochemical signal decreased when combined therapy was administered (Figure 2A,B). Regarding IDO-1 activity, there was a decrease in both regimens; however, it was statistically significant in R1 (Figure 2C). Similarly, immunohistochemical signals in neutrophil recruitment decreased when combined therapy was administered (Figure 2A,B).

### 3.2. Inhibition of IDO-1 and CXCR-2 Decreases Cell Proliferation

Furthermore, we also examined the role of combined therapy in cell proliferation. We performed an immunohistochemical assay for the proliferation markers MCM2 and PCNA. In cervical cancer tissue (Figure 3A,B, K14E7E2 panel), the PCNA and MCM2 stainings exhibited a nuclear pattern and were uniformly distributed throughout the thickness of the epithelium, consistent with previous reports on cervical cancer [41,42]. In the K14E7E2 + CT group, MCM2 staining was restricted primarily to basal and parabasal cells, indicating decreased proliferation. Regarding the PCNA marker, staining was limited to the basal and suprabasal zones (Figure 3A, K14E7E2 + CT panel; Appendix A).

### 3.3. Combined Therapy Targeting IDO-1 and CXCR-2 Increases Apoptosis and CD8+ Infiltration

IDO-1 or CXCR-2 have previously been reported to inhibit cancer cell apoptosis [43,44]. In the present study, we report an increase in apoptosis through the combined inhibition (CT) of IDO-1 and CXCR-2 in R1 (Figure 4A,C, tunel panel) and R2 (Figure 4B,D, tunel panel) regimens compared to K14E7E2 mice treated with vehicle.

The quantity and functionality of CD8+ T-cell tumor infiltration correlate with increased immunotherapy effectiveness [45]. To evaluate the efficacy of CT, we performed immunohistochemical staining of CD8+ and Granzyme B. The results showed increased CD8 and Granzyme B-positive cells when CT was administered in both regimens (Figure 4D,E).

### 3.4. Analysis of the Expression and Association of the IDO1 and CXCL Genes with Cervical Cancer Survival

To investigate the mRNA expression of the *IDO1* and *CXCL* genes of human cervical cancer, we performed an analysis based on the cancer genome atlas (TCGA) and genotype tissue expression (GTEx) data using the GEPIA2 platform. The data showed a significant increase in *IDO1*, *CXCL1*, and *CXCL8* mRNA expression in cervical squamous cell carcinoma and endocervical adenocarcinoma (CECS) (Figure 5A). A Kaplan–Meier analysis showed that *CXCL1* and *CXCL8* are prognostic and high expression is unfavorable in CC (Figure 5B).

## 4. Discussion

Elevated enzyme IDO-1 and dysregulation of the CXCR-2/CXCL axis have been reported in different tumors. They have been associated with an advanced stage of the disease, a poor prognosis, and an inadequate response to treatments [21,46,47,48,49,50,51]. For this reason, several reports suggest that inhibition of CXCR-2 or IDO-1 may be therapeutically helpful in many human cancers [52,53,54,55,56,57,58]. Regarding cervical cancer (CC), inhibition of IDO-1 (Navoximod) in combination with atezolizumab (PD-L1 inhibitor) has been studied in a phase 1 clinical trial (NCT02471846); however, antitumoral effectiveness was partial [36]. Recently, it was observed that IDO-1 inhibitors (D-1MT and DL-1MT) enhance the antitumor effect of the HPV16 E7 oncoprotein vaccine (gDE7) [59]. However, it is crucial to note that using IDO-1 inhibitors as a monotherapy approach promotes tumor growth [60]. Furthermore, Kenski et al. demonstrated that Epacadostat, an IDO-1 inhibitor, recently exhibited the ability to restore tryptophan levels. This, in turn, exerted an adverse tumor-protective effect on melanoma by preventing microphthalmia-associated transcription factor (*MITF*) downregulation [61]. In the present study, we observed a modest reduction in tumor burden when using the IDO-1 inhibitor DL-1MT as a monotherapy (Figure 1B, panel K14E7E2 + IDO1i).

Concerning CXCR-2, it has been reported to be essential in the development and persistence of CC [3,26,27]. Furthermore, inhibition of CXCR-2 in tumors with leukocytosis, including CC, has been suggested to be a promising therapeutic target [5,62]. However, the therapeutic effect of CXCR-2 inhibition in CC has been poorly studied. A study demonstrated that SB225002-treated HeLa and C33A cell lines decreased cell viability and induced cell apoptosis [24]. On the other hand, in different types of tumors, inhibition of CXCR-2 in combination with other therapeutic targets has greater antitumor activity [53,63,64]. Based on the information mentioned above, in this study, we reported a more significant antitumor effect when IDO-1i (DL-1MT) and CXCR-2i (SB225002) were administered in a combination regimen (Figure 1B, K14E7E2 + CT panel). Therefore, the combined inhibition of IDO-1 and CXCR-2 holds significant therapeutic value in CC compared to the monotherapy strategy.

Another crucial finding of this study is that the dual inhibition of IDO-1 and CXCR-2 led to decreased proliferation and increased apoptosis. These results are partially consistent with findings from both in vitro and in vivo studies. For instance, in clear cell renal carcinoma, chronic myelogenous leukemia, and lung cancer cell lines, SB225002 treatment induced apoptosis [53,65,66]. Additionally, CXCR-2 inhibition in the nasopharyngeal cancer cell line, suppression of proliferation, and induction of apoptosis were observed when inhibiting the MAPK pathway [67].

The antitumor capacity of IDO-1 inhibitors primarily stems from their regulation of immune cells, resulting in increased infiltration of tumor-infiltrating lymphocytes. In this study, we observed elevated levels of CD8+ T cells and an increase in Granzyme B expression consequent to DL-1MT and SB225002 treatment. Consistent with this finding, a combination of apo-IDO-1 inhibitor with apatinib [68] or sodium tanshinone IIA sulfonate (IDO-1 inhibitor) with anti-PD1 therapy [69] has been shown to increase CD8 + T-cell infiltration in colorectal cancer. Furthermore, DL-1 MT has demonstrated superior efficacy compared to L-1MT stereoisomers in lymphocyte proliferation and cytokine production [70]. It is important to note that when using IDO-1 inhibitors in CC therapy, the positivity rate of IDO-1 in this tumor varies between 52 and 100% [71]. Additionally, measuring the IDO-1 activity ratio should be considered, as this information has yet to be evaluated in some clinical trials (P1/2 ECHO-202/KEYNOTE-037 and P3 ECHO-301/KEYNOTE-252) [72,73].

In the K14E7 transgenic mouse model, E2 promotes progression through stages of CIN1, CIN2, CIN3, and CC. This model faithfully recapitulates the multistage process of human cervical carcinogenesis [37]. Additionally, estrogen in this model contributes to the sustained and malignant advancement of CC [40]. Furthermore, elevated estradiol concentrations have been documented within the tumor microenvironment (TME) of patients with CC [74]. In our study, we categorized experimental animals into two distinct treatment regimens based on their reliance on exogenous estradiol to sustain the growth of cervical tumors. Under the R1 regimen, the experimental cohort continued to receive E2 treatment, whereas the R2 regimen involved the discontinuation of E2 treatment. The findings in Figure 1 indicate a consistent efficacy of IDO-1 and CXCR-2 inhibition, irrespective of the tumor’s reliance on exogenous E2 treatment. This observation remains within the context of the combined therapeutic approach.

In the K14E7 and FvB models (control mice) treated with estradiol, we have reported an increase in *Ido1*, *Cxcls*, *CD274* (*PD-L1)*, and other inflammation-related genes [28]. Hence, we propose that estradiol is primarily responsible for deregulating these genes. Therefore, implementing the therapeutic approach utilized in this study for CC patients would necessitate subclassification based on the expression of these genes and prolonged hormonal stimulation, such as oral contraceptives, hormone replacement therapy, or pregnancy.

The study has a notable limitation concerning the absence of a comparison between the effects of PD-L1 + IDO-1 or PD-L1 + CXCR-2 inhibition with our therapeutic strategy. Nonetheless, this aspect holds promise for exploration in future experiments.

## 5. Conclusions

Our results revealed that the simultaneous inhibition of IDO-1 and CXCR-2 induced antitumor effects in cervical cancer. Therefore, our findings could have immediate clinical implications, considering that IDO-1 and CXCR2 inhibitors are already in clinical trials.

## Figures and Tables

**Figure 1 biomedicines-11-02280-f001:**
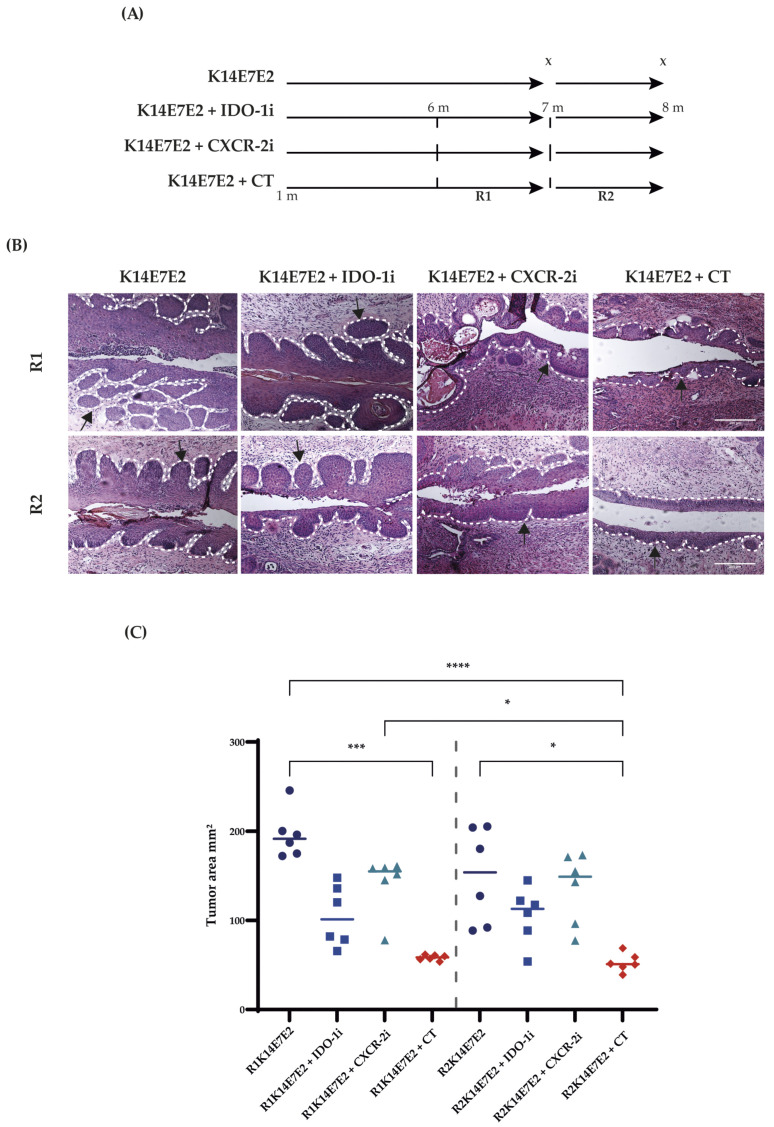
Combined inhibition of IDO-1 and CXCR-2 reduces the cervical tumor area. Figure (**A**) shows the schematic representation of R1 and R2 regimens. As described in the Materials and Methods Section, the R1 and R2 groups were treated with 1-DL-MT (IDO-1i) and SB225002 (CXCR-2i) for 30 days at 6 and 7 months of age and were sacrificed at the end of each regimen. The dotted lines represent the ages of the initiation of R1 and R2 regimens, while the x symbol represents the point of euthanasia. (**B**) Tissue sections were stained with hematoxylin and eosin, and histopathology was analyzed. The scale bar represents 100 µm, and the images were captured at 10× magnification. The arrows indicate the observed histological changes. (**C**) Tumor areas in cervical tissues were measured, and the Kruskal–Wallis and Dunn’s tests were used for statistical analysis (comparing matching R1 and R2). Geometric symbols represent individual mice, and the horizontal bars indicate the medians. CC: cervical cancer, IDO-1i: IDO-1 inhibitor; CXCR-2i: CXCR-2 inhibitor; CT: combined therapy, K14E7E2: mice treated with 17-beta estradiol for six months for the development of CC, 1, 6, 7 and 8 m: age of mice. * *p* < 0.05, *** *p* < 0.001 and **** *p* < 0.0001.

**Figure 2 biomedicines-11-02280-f002:**
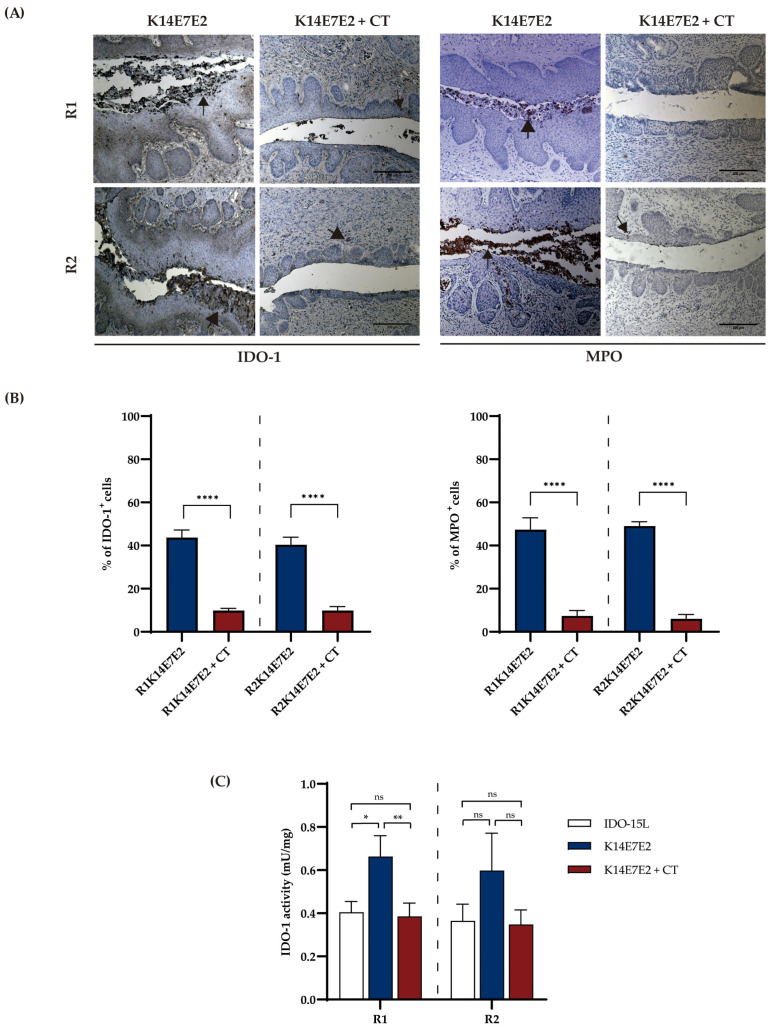
Combined therapy decreases the expression and activity of IDO-1 and neutrophil recruitment. (**A**) Immunohistochemical analysis of IDO-1 and MPO in cervical tissue from studied mice. The nuclei were counterstained with hematoxylin. The visual field at 10× magnification. Scale bar 200 µm. The arrows illustrate positive cells for IDO-1 and MPO. (**B**) Immunohistochemical-based quantification of IDO-1 and MPO. (**C**) Analysis of IDO-1 activity; pairwise comparisons were performed among all groups. IDO-15L was used as positive inhibition control in the IDO-1 activity assay. In (**B**,**C**), one-way ANOVA with Tukey tests was applied. In (**C**), Kruskal–Wallis with Dunn’s test was used. Each bar is a representative experiment from four independent assays. The analyses compared matching R1 and R2. * *p* < 0.05, ** *p* < 0.01, and **** *p* < 0.0001, ns = no statistical significance.

**Figure 3 biomedicines-11-02280-f003:**
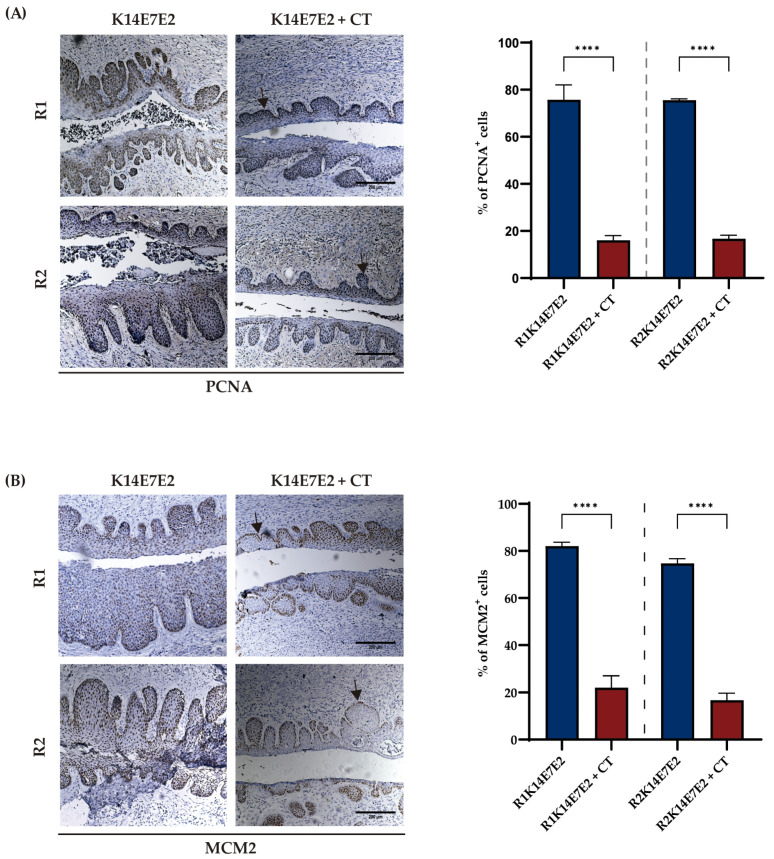
Combined therapy reduces the proliferation of cervical cancer tissue. The immunodetection pattern of PCNA (**A**) and MCM2 (**B**) shows a decrease in proliferation when combined therapy is administered (panel K14E7E2 + CT). The visual field at 10× magnification and scale bar 200 µm. The arrows illustrate positive cells for PCNA and MCM2. One-way ANOVA with Tukey tests was applied. Each bar is a representative experiment from four independent immunohistochemistry-based quantifications. The analyses compared matching R1 and R2 **** *p* < 0.0001.

**Figure 4 biomedicines-11-02280-f004:**
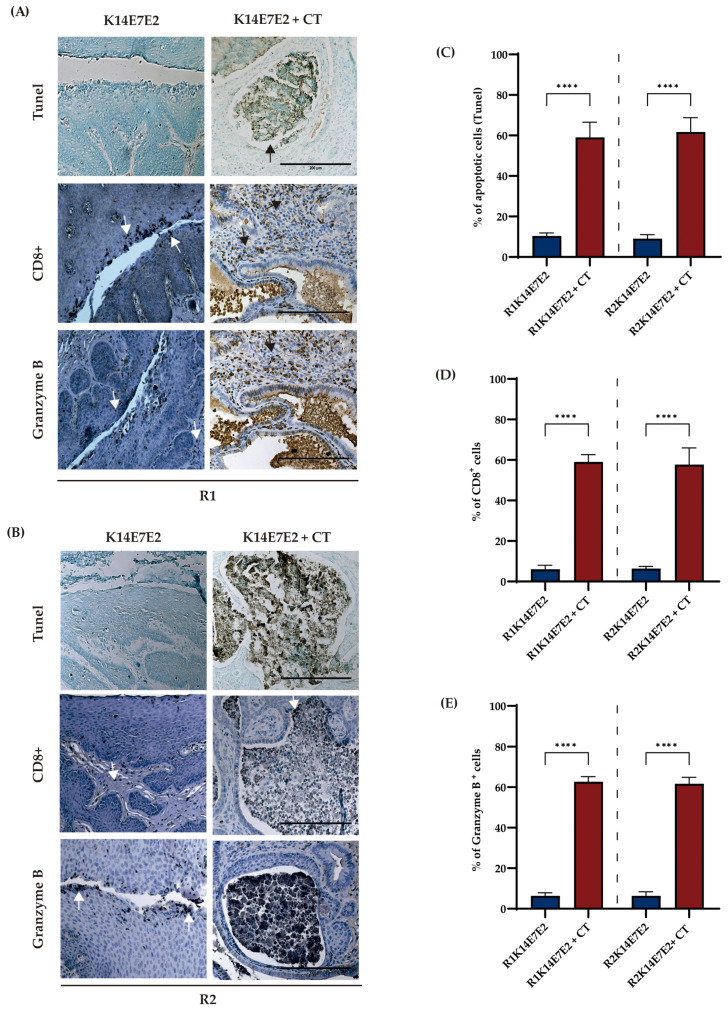
Combined therapy increases apoptosis and the number of CD8+ T cells in cervical tumors. (**A**,**B**) Schematic representation of tunel apoptosis assay, immunodetection of CD8+ T cells, and Granzyme B in R1 and R2 regimens, respectively. The visual field at 20× magnification and scale bar 200 µm. The arrows represent positive cells. Figures (**C**–**E**) illustrate the percentage of positive cells. One-way ANOVA with Tukey tests was applied. Each bar is a representative experiment from four independent immunohistochemistry-based quantifications. The analyses compared matching R1 and R2 **** *p* < 0.0001.

**Figure 5 biomedicines-11-02280-f005:**
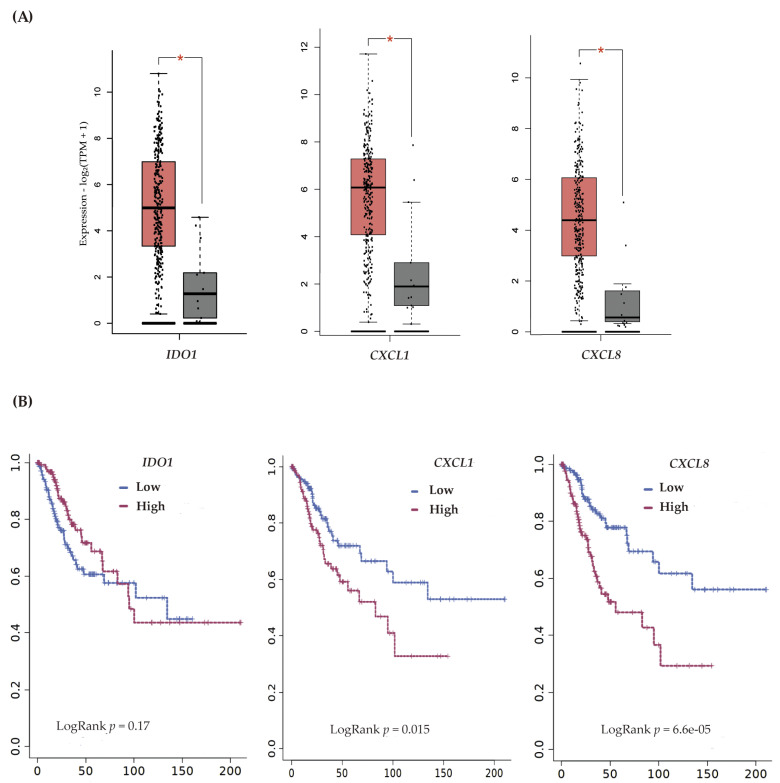
Expression of *IDO1*, *CXCL1*, and *CXCL8* and overall survival in human cervical cancer. (**A**) Analysis of mRNA expression and (**B**) overall survival in patients with cervical cancer according to low and high expression of *IDO1*, *CXCL1*, and *CXCL8*. In (**A**), Student’s *t*-tests, and Log2FC Cutoff > 1 and * *p* < 0.05 were applied. The boxplots depict CECS (in red) and normal tissue (in grey), with the medians represented by horizontal bars within the boxplots, and the whiskers delineate the ranges encompassing the lower 25% and upper 25% of the data values, excluding any outliers. In (**B**), a Kaplan–Meier curve was used.

## Data Availability

Not applicable.

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
