# Peer review of "Combined Inhibition of Indolamine-2,3-Dioxygenase 1 and C-X-C Chemokine Receptor Type 2 Exerts Antitumor Effects in a Preclinical Model of Cervical Cancer"

_biomedicines, 2023, doi:10.3390/biomedicines11082280_

Round 1

Reviewer 1 Report

The manuscript by Lizcano-Meneses et al is focused on understanding the importance of IDO1 and CXCR2 in mouse model of cancer. The authors have an appropriate in vivo model and defined the problem very well. The manuscript is acceptable for publication in its current form for publication.

Author Response

We appreciate the time spent reviewing our manuscript.

Reviewer 2 Report

The paper written by Solangy Lizcano-Meneses and collaborators concerns the combined effect of an IDO1 inhibitor with a CXCR2 inhibitor in 17-beta estradiol-induced cervical cancer in mice.

The article is quite complex, the authors determined a wide range of parameters to evaluate the effect of this type of combined therapy in cervical cancer

A few questions:
how was the amount of IDO1 inhibitor water ingested by each mouse determined? In the article it is specified that it is an average of 5 ml of water daily. I think it is important to specify this because it is the concentration of IDO1 administered.
What is the relevance of the treatment started at different times ? The authors divide the experimental animals into two treatment regimens; and all determinations are made comparing the 2 groups. However, in the discussion section, no comment is made on what was the overall impact (if any) of starting treatment one month later in the R2 group.

Author Response

We appreciate the time spent reviewing our manuscript; the observations and comments have been addressed.

how was the amount of IDO1 inhibitor water ingested by each mouse determined?

RESPONSE: This represents a study limitation. Initially, we attempted to administer the IDO1 inhibitor using oral gavage; however, some mice died before completing the treatment. Consequently, we opted to administer it via a cage drinker, and as indicated, this treatment method was successful.

In lines 109-111, we add, “To ascertain the average daily water consumption of the mice, we calculated the difference in solution volume and subsequently divided it by the number of days and the number of mice in each cage”.

Mittal D. et al. have previously reported this treatment strategy (reference number 38).

To clarify the relevance of the treatment started at different times. In lines 116-117, the following was written: “Due to exogenous estradiol dependency for the continued growth of cervical tumors, K14E7E2 mice were divided into two treatment regimens”.

In lines 415-426 of the discussion section, the following paragraph was added “In K14E7 transgenic mouse model, E2 promotes the progression through stages of CIN1, CIN2, CIN3, and CC. This model faithfully recapitulates the multistage process of human cervical carcinogenesis [31]. Additionally, estrogen in this model contributes to the sustained and malignant advancement of CC [34]. Furthermore, elevated estradiol concentrations have been documented within the tumor microenvironment (TME) of patients with CC [68]. In our study, we categorized the experimental animals into two distinct treatment regimens based on their reliance on exogenous estradiol to sustain the growth of cervical tumors. Under the R1 regimen, the experimental cohort continued to receive E2 treatment, whereas the R2 regimen involved the discontinuation of E2 treatment. The findings in Figure 1 indicate a consistent efficacy of IDO1 and CXCR2 inhibition, irrespective of the tumor's reliance on exogenous E2 treatment. This observation remained within the context of the combined therapeutic approach.

Reviewer 3 Report

The study demonstrates that IDO1 and CXCR2 have anti-tumor effects in a preclinical model of cervical cancer.

Authors established a K14E7E2 transgenic mouse mode to investigate the effects of IDO1 and CXCR2 in cervical cancer.

A reason to investigate a combination of 1-DL-MT and SB225002 may be explained more in detail, preferably in Introduction.

The way of describing p value in Figures may be checked.

Author Response

We appreciate the time spent reviewing our manuscript; the observations and comments have been addressed.

RESPONSE: To clarify the relevance of investigating a combination of 1-DL-MT and SB225002 in CC, the following paragraph was edited:

Lines 68-87 “Immunotherapy targeting TIME has emerged as a promising approach. In the context of CC, the FDA has approved using pembrolizumab in patients who test positive for programmed cell death ligand 1 (PDL-1). However, the response rate to this treatment has been relatively low, ranging from 10% to 17% [29,30]. Single-agent immunotherapies, exemplified by agents like pembrolizumab in the context of CC, may encounter limitations due to tumor heterogeneity, immune escape mechanisms, or microenvironmental factors [31]. Combining different immunotherapeutic agents, such as immune checkpoint inhibitors, adoptive T cell therapies, cytokines, and vaccines, can be created a more comprehensive and dynamic immune response against the dis-ease. For example, studies indicated that when used as a single agent, IDO1 or CXCR2 inhibitors exerted little antitumor efficacy, while combination with other therapies showed markedly enhanced antitumor effectiveness [32-35]. On the contrary, a phase 1 clinical study yielded inconclusive evidence concerning the potential benefits of combined therapy incorporating the IDO1 inhibitor (navoximod) and the PD-L1 inhibitor (atezolizumab) in solid tumors, including CC. [36]. Therefore, exploring the effects of IDO1 inhibitor (1-DL-MT) and CXCR2 inhibitor (SB225002) can be useful by their distinct mechanisms of action, which hold the potential for a complementary impact on specific pathways crucial to the progression of CC. Our study aimed to investigate the influence of inhibiting IDO1 and CXCR2 on cervical tumor growth. To achieve this, we utilized pharmacological inhibitors in single-drug or combination approaches using a preclinical model of cervical cancer (K14E7E2)”.

All p values were described as follows:  *p < 0.05, **p < 0.01, ***p < 0.001 and ****p < 0.0001.

Reviewer 4 Report

This is a well written paper by Lizcano-Meneses and co-workers evaluating the effects of combined IDO1 and CXCR2 inhibition on cervical cancer proliferation and tumor immune microenvironment in preclinical mouse model.

Authors presented results suggest that targeting IDO1 and CXCR2 inhibits cervical cancer proliferation and reduces cancer area in mice. Also tumor tissue is enriched with CD8+ cells. Finally, authors concluded that simultaneous inhibition of IDO1 and CXCR2 may serve as novel neoadjuvant modality in anti-cervical cancer therapy.

This is an interesting study, and clinically valuable. The idea and results presented in this manuscript may attract attention of some groups of researchers, especially those searching for new anti-cancer modalities.

There are few minor issues that need to be addressed:

Comment 1. Following Combined Therapy (CT) proliferation marker MCM2 (% of MCM2 positive cells) decreases significantly in Fig. 3B while it increases significantly in Fig. 4A. It is quite confusing. Furthermore, authors do not refer to these results in the text (Results 3.3 section).

Comment 2. Figure 5A should be described in detail, what red and grey bars represent.

 Taken together, this paper by Lizcano-Meneses and collegues represent a worthwhile contribution to the cancer research. I recommend the manuscript for further publication.

Author Response

We appreciate the time spent reviewing our manuscript; the observations and comments have been addressed.

RESPONSE: There is an error in the legend of the vertical axis in Figure 4C. The accurate axis legend should be "% of apoptotic cells (TUNEL)" instead of "% of MCM2 cells." We trust that now the findings in Figure 4 are clarified.

Lines 363-365 describe described what red and grey bars represent. "The boxplots depict CECS (in red) and normal tissue (in grey), with the medians represented by horizontal bars within the boxplots and the whiskers delineate the ranges encompassing the lower 25% and upper 25% of the data values, excluding any outliers"